# Structure-Function of the Human WAC Protein in GABAergic Neurons: Towards an Understanding of Autosomal Dominant DeSanto–Shinawi Syndrome

**DOI:** 10.3390/biology12040589

**Published:** 2023-04-12

**Authors:** Hannah C. Rudolph, April M. Stafford, Hye-Eun Hwang, Cheol-Hee Kim, Jeremy W. Prokop, Daniel Vogt

**Affiliations:** 1Department of Pediatrics and Human Development, Michigan State University, Grand Rapids, MI 49503, USA; 2Department of Biology, Chungnam National University, Daejeon 34134, Republic of Korea; 3Office of Research, Corewell Health, Grand Rapids, MI 49503, USA; 4Neuroscience Program, Michigan State University, East Lansing, MI 48824, USA

**Keywords:** cell biology, protein sorting, nuclear translocation, protein domain, WAC

## Abstract

**Simple Summary:**

There are several rare, disrupted genes that underlie neurological dysfunction. Many have not been characterized for their role in brain function or how they work in individual brain cells. The WW domain-containing adaptor with coiled-coil, *WAC*, gene is one example. Dysfunction of this gene underlies a rare syndrome, DeSanto–Shinawi syndrome (DESSH), with those diagnosed having symptoms including cranio-facial changes, autism, and attention deficit hyperactivity disorder. We sought to understand how the WAC protein functions in brain cells implicated in DESSH in two ways. First, we used available technologies to predict important conserved regions in the protein that may underlie cellular function, including how it localizes to distinct areas of a cell, and further correlated these findings with reported human genetic variants in these regions. These efforts uncovered novel regions in the protein necessary and sufficient for it to localize to the nucleus. Second, we deleted/used key regions of the WAC protein to test whether they were necessary/sufficient to localize WAC to distinct cell regions in brain neurons, and we found that the amino-terminus of the protein fulfilled this function. Moreover, other regions contribute to distinct biological functions of WAC, and this study first highlights these aspects of this unique neurodevelopmental protein.

**Abstract:**

Dysfunction of the WW domain-containing adaptor with coiled-coil, *WAC*, gene underlies a rare autosomal dominant disorder, DeSanto–Shinawi syndrome (DESSH). DESSH is associated with facial dysmorphia, hypotonia, and cognitive alterations, including attention deficit hyperactivity disorder and autism. How the WAC protein localizes and functions in neural cells is critical to understanding its role during development. To understand the genotype–phenotype role of WAC, we developed a knowledgebase of WAC expression, evolution, human genomics, and structural/motif analysis combined with human protein domain deletions to assess how conserved domains guide cellular distribution. Then, we assessed localization in a cell type implicated in DESSH, cortical GABAergic neurons. WAC contains conserved charged amino acids, phosphorylation signals, and enriched nuclear motifs, suggesting a role in cellular signaling and gene transcription. Human DESSH variants are found within these regions. We also discovered and tested a nuclear localization domain that impacts the cellular distribution of the protein. These data provide new insights into the potential roles of this critical developmental gene, establishing a platform to assess further translational studies, including the screening of missense genetic variants in *WAC*. Moreover, these studies are essential for understanding the role of human *WAC* variants in more diverse neurological phenotypes, including autism spectrum disorder.

## 1. Introduction

Rare disease syndrome genes have increasingly been identified as an underlying cause of neuropsychiatric and neurological symptoms, with more than one thousand genes suggested to contribute to neurodevelopment [1]. The protein products of these genes work together at the synapse or during cellular signaling, or they have a role in epigenetic and transcriptional regulation [2]. However, some gene products have not yet been functionally assessed to determine cellular roles, so understanding of how they impact neurodevelopment is lacking, preventing potential therapeutic interventions. Having approaches to characterize these genes and their protein products is critical for advancing sequencing into clinical utility [3]. Here, we provide an example of developing screening tools for an underexplored neurodevelopment gene, elucidating novel molecular functions that could impact neural cell types.

The WW domain containing adaptor with coiled-coil (*WAC*) gene is associated with DeSanto–Shinawi syndrome (DESSH). Those diagnosed with DESSH have symptoms of cranial dysmorphia and hypotonia with comorbidities including attention deficit hyperactivity disorder (ADHD), autism, and seizure susceptibility [4,5,6,7,8]. Moreover, *WAC* variants rank it as a high-confidence autism spectrum disorder risk gene [9,10]. The WAC protein has a few characteristic conserved domains, yet most of the protein is structurally disordered and not yet characterized. Notably, the WAC protein is often enriched in the nuclei of cells [11,12], but no nuclear localization signal (NLS) has been defined in the protein due to no sequences matching classical NLS signals. While implicated in cellular signaling and nuclear regulation [12,13,14,15], it is unclear how regions of the WAC protein modulate cellular distribution and how human variants could alter these functions to impact critical brain cell types. Therefore, we applied a computational screening combined with cell assays to define further the domains and disorder motifs of the WAC protein, serving as an example of characterizing underexplored neurodevelopment genes.

To ascertain the structure-function of WAC in a model cell type, we examined the cellular distribution of the protein in cells recently implicated in WAC phenotypes, cortical GABAergic interneurons [16], with the rationale that cell types sensitive to WAC mutations would have the most translatable value. Thus, we chose to utilize mouse medial ganglionic eminence (MGE) cells, which give rise to the majority of GABAergic cortical interneurons [17], as a model system to examine the impact of WAC protein domain mutants on cellular distribution. This approach offers a unique ability to screen the functional regions of a protein in a relevant neural cell type, a critical need for the field of neurodevelopmental genetics. Our analyses uncovered a unique bipartite nuclear localization signal in the amino terminus with an additional C-terminal nuclear export signal. Notably, these regions are enriched for human clinical variants. This finding suggests that nuclear localization is critical for proper WAC function, and its dysfunction results in DESSH. Further, a large number of conserved motifs within the disordered regions of the protein, where clinical variants fall, suggest that posttranslational modifications and cellular signaling are influential aspects of WAC function.

## 2. Materials and Methods

### 2.1. Bioinformatics

The bioinformatics of WAC were assessed using a combination of previously published tools [18,19]. In short, sequences of vertebrate WAC protein were extracted from NCBI Gene Ortholog [20] during October 2022 and aligned with NCBI Cobalt [21]. UniProt annotations [22], gnomAD version 2 variants [23], ClinVar variants [24], and Geno2MP (http://geno2mp.gs.washington.edu, accessed on October 2022) variants were extracted during November 2022. The WAC protein structure came from AlphaFold prediction [25]. The human WAC protein sequence (UniProt sequence Q9BTA9, Ensembl transcript ENST00000354911) was assessed for linear motifs through the Eukaryotic Linear Motif (ELM) server [26]. *WAC* gene expression was annotated from GTEx human broad tissues, the Allen Brain Atlas BrainSpan [27] of different ages vs. regions of the human brain, and the Allen Brain Atlas Human MTG 10X (SEA-AD) single cell dataset [28]. A combined list of all details of the amino acids of WAC can be found at https://doi.org/10.6084/m9.figshare.22263538.v1 (accessed on October 2022).

### 2.2. Animals

A colony of wild type (WT) CD-1 mice were bred to generate the embryos used for primary cultures. Experimenters were blinded to the parameters. All mouse procedures were performed in accordance with NIH Guidelines for the Care and Use of Laboratory Animals and were approved by the Michigan State University Institutional Animal Care and Use Committee.

### 2.3. DNA Vector Generation

The *CMV-GFP-hWAC* DNA vector was generated by amplifying human *WAC* cDNA from a DNASU plasmid (hsCD00442491) and cloning in frame to the 3′ end of GFP in a previously described GFP expressing plasmid [29]; *WAC* was cloned into 5′ BsrGI and 3′ BamHI restriction enzymes sites. We subsequently edited a 3′ deletion of two adenosines in the *WAC* coding domain to produce the correct full-length clone. The following primers were used to generate the full length and mutants: WAC forward 5′-GAGATGTACAAGATGGTAATGTATGCGAGG-3′, and WAC reverse 5′-GAGAGGATCCTCACACCATGAAGGAATTC-3′. Forward primers contained a BsrGI restriction site and reverse primers a BamHI restriction site (underlined); numbers denote amino acids of WAC.

W90 forward 5′-GAGATGTACATTGATGGTGGGACCAGTTAC-3′

W166 forward 5′-GAGATGTACATTGAACAGAGACAAAAAGAAGC-3′

W576 forward 5′-GAGATGTACATTGATCATGCAGAGAAGCAG’3′

W89 reverse 5′-GAGAGGATCCTCACCTCTCTCTAACTCTGTG-3′

W165 reverse 5′-GAGAGGATCCTCATCTTTCAAGCCACTCTTTTGG-3′

W575 reverse 5′-GAGAGGATCCTCATGCAGGCCATCCTTGAAC-3′

The WW domain deletion construct was generated by obtaining a synthesized gene block from integrated DNA technologies (IDTs) of a partial WAC coding domain that lacked the WW DNA sequence (encoding amino acids 133–165) with flanking 5′ BsrGI and internal 3′ NsiI restriction sites that were used to ligate in the GFP-WAC vector described above.

### 2.4. Immuno-Fluorescence Labeling and Imaging

Primary neurons were washed in phospho-buffered saline containing 0.3% Triton-X100, blocked in the same solution containing 5% bovine serum albumin and then incubated in primary antibodies for 1–2 h. They were washed three times and then incubated with secondary antibodies containing fluorophores for 1 h before three final washes. A Leica DM2000 microscope (DFC3000G camera) captured primary cell images for quantification, and a Zeiss LSM800 confocal microscope was used to attain representative images for figures. A blinded individual scored the nuclei/cytoplasm and obtained punctate measurements by counting GFP+ cells merged with DAPI; to be nuclear or cytoplasmic, the majority of GFP signal had to reside in that compartment, and punctate status was determined if any punctate spots were observed.

### 2.5. MGE Primary Cultures

We generated primary cultures from embryonic day (E) 13.5 MGE tissue as described in [30,31]. Briefly, MGE tissue was dissected, triturated to mechanically dissociate into single cells, and then seeded and cultured in Dulbecco’s modified Eagle medium (DMEM) supplemented with 10% fetal bovine serum (FBS) and penicillin/streptomycin from the time of seeding until one day in vitro. The cells were then transfected with the GFP-fusion vectors using Lipofectamine^TM^ 2000 (ThermoFisher 11668027) on day one and, after four hours, replaced with neurobasal media, supplemented with B27, glucose, glutamax, and penicillin/streptomycin. Cells grew in this medium for five days in vitro and then w2ere fixed in 4% paraformaldehyde and assessed via immuno-fluorescence labeling.

### 2.6. Western Blotting

HEK293T cells, cultured in DMEM with 10% FBS, were transfected with DNA plasmids using Lipofectamine^TM^ 2000. After 48 h, cells were collected and lysed in standard RIPA buffer with protease and phosphatase inhibitors and combined with Laemmli buffer (BioRad 1610737EDU) containing 2-mercaptoethanol and incubated at 95 °C for 5 min. Equal amounts of protein lysates were separated on 10% SDS-PAGE gels and then transferred to nitrocellulose membranes. The membranes were washed in Tris-buffered saline containing Triton X-100 (TBST) and blocked for 1 h in TBST containing 5% non-fat dry milk (blotto, sc-2324 SantaCruz biotechnology). Membranes were incubated with primary antibodies overnight at 4 °C, washed three times with TBST, incubated with secondary antibodies for 1 h at room temperature, and then washed three more times with TBST. Membranes were incubated in ECL solution (BioRad Clarity substrate 1705061) for 5 min, and chemiluminescent images were obtained with a BioRad Chemidoc™ MP imaging system. Antibodies (all used at 1:4000 dilution): rabbit anti-GFP (ThermoFisher A6455), rabbit anti-WAC (abCam ab109486), and rabbit anti-GAPDH (Cell Signaling Technology 2118), as well as goat anti-rabbit HRP (BioRad 170-6515).

### 2.7. Statistics and Cell Assessments

Graphpad Prism software, version 7, was used to calculate statistical significance; a *p* value of <0.05 was considered significant. For non-parametric data sets (proportions), we used the chi-square test to determine significance. GFP distribution and punctate analyses sampled 20–25 cells per primary culture, from three independent biological replicates.

## 3. Results

### 3.1. Functional Domain and Motif Annotations of WAC

The WAC protein contains an internal WW domain, known for protein–protein interactions with proline rich regions [32]. At the carboxy-terminus, the WAC protein has two coiled-coil domains, which can have many functions, including dimerizing with similar structures [33]. Between these regions of the WAC protein are highly disordered protein annotations with no known functions. Using 364 vertebrate WAC protein sequences, we identified regions with high conservation (Figure 1). While UniProt only annotates the WW domain and coiled-coil region of WAC with the majority predicted to be disordered, there is a surprising high level of conserved polar basic (blue, top panel Figure 1A), polar acidic (red, top panel Figure 1A), and serine/threonine (orange, top panel Figure 1A) conservation. Most of these conserved sites are found outside of the WW and coiled-coil annotations, suggesting an under-annotated role of disordered motifs in WAC.

According to the Human Protein Atlas (https://www.proteinatlas.org/ENSG00000095787-WAC/subcellular, accessed on October 2022), WAC is localized within the nucleus of cells but lacks an annotated nuclear localization domain. We screened the WAC sequence with multiple NLS bioinformatic screening tools, without a single tool returning a potential NLS. Surprisingly, a large portion of the linear motif predictions of WAC are also identified to be within the nuclear cell compartment, suggesting sequence convergence to nuclear function. These nuclear motifs include conserved sites for DOC_MAPK_gen_1 (169-178, 606-614), LIG_FHA_2 (307-313, 618-624), DEG_SPOP_SBC_1 (322-326), DOC_CYCLIN_yClb5_NLxxxL_5 (357-366), DEG_APCC_DBOX_1 (363-371), DOC_CYCLIN_yCln2_LP_2 (366-372), DOC_PP2A_B56_1 (383-388, 611-617), LIG_PCNA_PIPBox_1 (400-409), LIG_EH1_1 (413-421), LIG_CtBP_PxDLS_1 (437-441), LIG_14-3-3_CanoR_1 (454-458, 517-526), DOC_CKS1_1 (455-460, 469-474), DEG_SCF_FBW7_1 (468-475), DOC_CKS1_1 (469-474), DOC_PP1_RVXF_1 (623-629), and LIG_UBA3_1 (626-634). Each of the above motifs has an included hyperlink to descriptions within the ELM database. It should be noted that several of these regulation motifs are linked to similar neurodevelopmental disorders including MAPK dysfunction and DEG_SCF_FBW7_1, which was observed as modified in multiple individuals with MED13-related neurodevelopmental disorder [34]. In addition, DEG_SPOP_SBC_1 suggests that the protein localizes to nuclear puncta via the Cul3-RING ubiquitin ligase complex, and multiple motifs suggest the protein has cell cycle and DNA damage impacts through destruction motifs and phospho signaling.

Polar basic amino acids are often critical for nuclear localization, with WAC having the highest density of conserved polar basic residues from amino acids 77-85. Polar basic amino acids 57, 60, 61, 68, 75, 79, 82, 84, 85, 87, and 89 are conserved in greater than 90% of sequences analyzed (DPSPPN**K**ML**RR**SDSPEN**K**YSDSTG**H**SKA**K**NV**H**T**HR**V**R**E**R**DGGTSYSPQENSHNHSALHSS). Near this region is an ELM annotated MOD_SUMO_rev_2 site (amino acids 62-70), a SUMOylation modification site found within nuclear proteins. Additionally, a TRG_NES_CRM1_1 nuclear export signal (NES) is found within amino acids 604-616, flanked by a MOD_SUMO_rev_2 site (amino acids 599-608). This finding suggests that nuclear localization is potentially regulated by both the N- and C-terminal regions through nonclassical motifs.

Many of the conserved amino acids of WAC fall outside the annotated domains/motifs within unstructured regions of WAC. Structure predictions show that several of the most conserved linear motifs are found in unstructured regions of the AlphaFold predicted structure (Figure 2), in line with potential functional, intrinsically disordered regions of WAC.

### 3.2. Human Clinical Variants Annotated to Functional Motifs of WAC

There are multiple rare disease variants that fall within the newly identified motifs of WAC (Figure 1B). Within ClinVar, there are loss-of-function (LoF) variants, suggesting that truncation of the protein results in disease. While few ClinVar missense variants have been identified to date, they may also underlie symptoms of DESSH. A large number of clinical phenotypes for rare disease patients can be observed for variants within the Geno2MP database. There is an enrichment for nervous system abnormalities (Table 1) of clinical variants with annotated phenotypes from Geno2MP predicted to have high pathogenic CADD scores (>20). We further analyzed three regions with clinical variants within functional motifs (Figure 3).

Within and adjacent to the possible polar basic N-terminal region of WAC that may regulate nuclear localization, five variants are found at conserved sites in Geno2MP. Two variants, S94I and Y95C, are observed multiple times in affected individuals with nervous system phenotypes. This site is a potential phosphorylation switch. The S94I variant was found homozygous in three individuals, with plastic paraplegia, abnormality of brain morphology, and intellectual disability. One individual heterozygous for the variant exhibited seizures. The Y95C variant was heterozygous in one affected individual with encephalocele. It should be noted that none of the polar basic amino acids observed in Geno2MP individuals varied.

A second conserved motif of WAC (510–534) contains multiple phosphorylation sites and a 14-3-3 interaction motif, a site that requires phosphorylation for interactions to occur. The critical amino acid R517 for 14-3-3 interaction has two variants within Geno2MP connected to neurological phenotypes. The R517C variant was found heterozygous in two affected individuals with epileptic encephalopathy, and R517H was found heterozygous in an individual with neurodevelopmental abnormality. 14-3-3 signaling has been suggested to have a neuroprotective function with elevation in seizure model systems [35].

The most surprising motif with functional variants connected to neurological phenotypes in Geno2MP is the C-terminal region with the nuclear export signal (WAC 596–636). This region, in addition to the nuclear export signal, has a SUMOylation motif, several phosphorylation sites, and multiple protein interaction sites. None of the Geno2MP variants are found in the nuclear export signal; rather, they flank this site. E600K is found within the SUMOylation motif and in two individuals with nervous system abnormalities. The other variants fall within a DOC_PP1_RVXF_1 motif, which is a dephosphorylation site. R625S was found homozygous in three affected individuals, including one with seizures and one with abnormality of the nervous system. Four affected individuals were heterozygous for R625S, with two having abnormality of brain morphology and one with intellectual disability. I626L is the most common homozygous Geno2MP variant, with 16 affected individuals annotated with abnormal brain morphology, four with spastic paraplegia, and three with intellectual disability. The variant is absent in the gnomAD and TOPmed databases, suggesting significant enrichment in neurological disease. L627I was found heterozygous in two affected individuals with spastic paraplegia, and Q632K was found heterozygous in four affected individuals with abnormality of brain morphology. Overall, this outcome suggests that N and C-terminal regions with nuclear localization and export signals are critical for neurological development and associated with human variants of neurological phenotypes.

### 3.3. Expression of WAC within Human Brain and Neural Cell Types

To identify cell types that WAC may impact, we performed an analysis of expression datasets (Figure 4). In the GTEx database of broad human tissue expression, several protein isoforms are observed to be expressed for WAC (Figure 4A). The longest isoform codes for a 647 amino acid protein, with highest expression in all tissues and enriched within the brain. Smaller isoforms with spliced exons (602 and 544 amino acids) are also highly expressed in the brain and have all three of the annotated domains (Figure 4B). The BrainSpan human developmental brain atlas shows that *WAC* expression is relatively the same throughout post-coital weeks (pcw), with higher variability later in life (Figure 4C). This outcome suggests that, within the developmental window (pcw8-37) and in the early months of life, the expression of WAC is finely controlled, similar to many other genes involved in neurological development. Multiple regions of the brain also show high expression of WAC (Figure 4D), suggesting that broad neural classes are of interest to WAC expression within development. From human single cell brain expression, both GABAergic and glutamatergic neurons express *WAC*, while most non-neuronal cell types have significantly lower expression (Figure 4E). These data suggest that the broad GABAergic and glutamatergic neurons found throughout brain regions have a high level of WAC expression and that expression is well controlled in the early developmental stages. However, no studies to date have addressed the role of WAC and its nuclear localization in these neural cell types.

### 3.4. Generation and Structure/Function of GFP-WAC Mutant Fusion Proteins

To assess the contribution of WAC protein domains to cellular localization and subcellular distribution, we generated a series of mutant WAC-expressing plasmids that included an amino-terminal GFP fusion; amino-terminal fusion of a different epitope was used to study WAC protein localization [11], suggesting that this approach would be feasible. The plasmids included protein domain deletions to assess whether specific regions were necessary and/or sufficient for nuclear localization and protein domain fusions to test sufficiency to localize in neurons (schema of plasmids, Figure 5A).

We first assessed whether these GFP-WAC expression plasmids generated protein products of the predicted molecular weight via western blot. These assays assessed the GFP fusion proteins and whether the same mutant proteins were recognized by a functional commercial antibody raised against WAC. To this end, we expressed the plasmids in HEK293T cells and collected protein lysates at 48 h for western blot analyses (Figure 5B). Notably, each of the expressed proteins exhibited GFP+ bands via western blot at the expected sizes (Figure 5C, upper panel), suggesting that these fusion proteins could be properly expressed, and the deletion mutants were stable enough to be detected via this method.

We also used a commercial (abCam ab109486) antibody raised against the WAC protein on the deletion mutants to detect where the epitope may reside in WAC (Figure 5C, lower panel). This antibody only detected protein when amino acids 166-575 were present, suggesting the antibody epitope resides within this region. The faint bands found in all lanes are likely the endogenous WAC protein isoforms expressed by the HEK293T cells. Overall, these data demonstrated that we could express the various WAC mutant proteins and determine the epitope region used to develop a reliable commercial WAC antibody.

### 3.5. WAC Deletion Mutant Proteins Exhibit Differential Targeting and Aggregation in Neurons

We expressed the GFP-WAC full-length gene and mutant versions in brain-derived MGE cells. We collected and cultured embryonic day (E)13.5 MGE cells. The cells were transfected with the GFP fusions at 24 h and allowed to develop for five days, a time when their morphology would allow us to decipher cytoplasmic vs. nuclear localization of the proteins. To this end, we found multiple phenotypes related to the necessity or sufficiency of these protein domains related to nuclear/cytoplasmic localization, as well as distinct sequestration/puncta-like distribution in neurons expressing each mutant.

Consistent with a previous study [11], full length GFP-WAC preferentially localized to the nuclei in MGE cells (Figure 6A,A’,I). To determine the role of distinct domains in cellular localization and distribution, we quantified whether various GFP-WAC deletion proteins localized in the nucleus and/or cytoplasm, as well as in discrete puncta within the neurons. Importantly, all mutant proteins that contained the N-terminal region localized preferentially to the nucleus (Figure 6B–E,B’–E’,I), suggesting that the predicted NLS in this region may be functional. Notably, that region alone was sufficient to localize GFP to the nucleus (Figure 6B,B’,I). In agreement with this observation, the proteins that lacked the N-terminal region were found in the cytoplasm (Figure 6F–H,F’–H’,I), suggesting that nuclear sequestration of WAC is dependent on the N-terminal basic amino acid-enriched region.

In addition to nuclear sequestration of WAC, we also noted the puncta appearance of the protein when certain domains were present. For example, the full-length protein and WW deletion mutant both have a distinct punctate pattern (Figure 6A,A’,E,E’,J), suggesting that the WW domain is dispensable for this phenotype. Consistent with the WW domain having no impact on the punctate distribution of WAC, neither the putative NLS nor the CC domains seemed to have any influence over this pattern. The region of the protein that may regulate this punctate distribution likely resides between amino acids 166 and 575 (Figure 6J), the region predicted earlier to contain several linear motifs with potential phospho targets. One caveat is that some distribution could also be due to degradation of the proteins that cannot be excluded here. Thus, while we have determined the contribution of many of WAC’s conserved domains on cellular localization, there is still more to be learned about this unique syndromic protein.

## 4. Discussion

Dysfunction of the WAC gene underlies DESSH, as well as several comorbid neurological symptoms, including autism spectrum disorder, ADHD, and seizures [4,5,7,7,8]. This dysfunction likely arises from genetic variants within key conserved regions of the WAC gene and/or loss of function; some of these observations have elevated WAC to be considered a high-confidence autism risk gene [9,10], conveying the importance of understanding how the gene, protein, and downstream effectors are changed, as well as potential inroads into therapeutics to remedy these alterations. Our structure-function approach begins to delve into how these regions of the WAC protein may be involved in these processes and provide a template for future work examining the roles of genetic variants and other phenotypic outcomes that *WAC* underlies.

The WAC protein has several conserved domains and unexplored regions. The WW and CC domains have been reported in previous reports [11,14], but how WAC localizes to the nucleus and whether other regions may dictate functions have yet to be addressed. Herein, we found that the amino terminus of WAC was enriched for basic amino acids, and this region was both necessary and sufficient for nuclear sequestration. Defining the region in WAC for nuclear localization will be crucial moving forward, as human genetic variants in this region are assessed. In addition, the WAC protein has a punctate pattern in neurons, suggesting some type of localization to cellular compartments. This finding was observed in the nucleus and cytoplasm of various deletion mutants, suggesting that some part of the WAC protein can mediate this observation. While future studies are needed to assess how this punctate pattern manifests, our data suggest that amino acids 166-575 may be responsible for this cellular pattern. This region is highly conserved but lacks the conventional known conserved WW and CC domains. Instead, this region is mainly linear in nature and contains several amino acids that could act as phospho-switches. Interestingly, this finding adds another level of regulation to this important protein and suggests that other cellular events may be involved in WAC functions.

Our assessments herein focused on GABAergic neurons due to our previous data, suggesting that some of these neurons were susceptible to *Wac* loss of function [16]. We also performed an assessment of *Wac* expression in brain cell types and found a preferential enrichment for both GABAergic and glutamatergic neurons. Unfortunately, *Wac* deletion is embryonically lethal, but conditional mutants may be able to uncover further functions of this gene in both GABAergic and glutamatergic neurons in future studies. The role of WAC’s conserved regions reported here is a strong starting point in understanding the various mechanisms by which this protein regulates each neuron subtype. We predict that, while the localization of this protein may be similar between cell types, the functional partners may differ; thus, the phenotypes may be distinct within each brain cell type. To this end, we found that the same WAC mutants used in MGE primary neurons had the same distribution and punctate patterns in HEK293T cells (HCR and DV, data not shown).

*Wac* is an important developmental gene with relatively few known functions. Herein, we developed the first assessment of this important protein’s conserved regions in cellular localization. These studies will help to interpret the growing number of genetic variants in *Wac* associated with neurodevelopmental disorders, including autism [9,10]. In addition, we found unique roles for uncharacterized regions in the WAC protein that will be assessed in future studies, which may provide unique roles for how WAC operates and elucidate the cellular pathways that utilize WAC for various functions.

## Figures and Tables

**Figure 1 biology-12-00589-f001:**
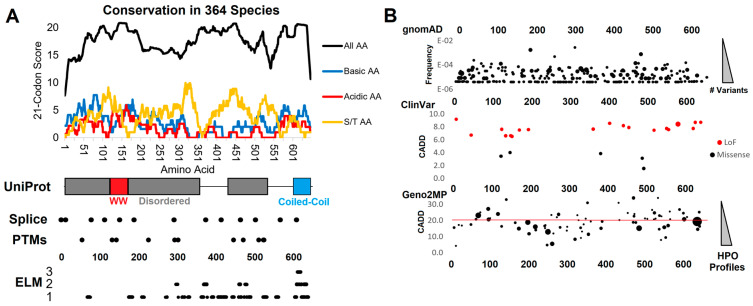
**Genomic analysis of WAC**. (**A**) Compiled genomics insights for WAC. On top is the conservation of 364 species of WAC protein. The conservation score of all amino acids (black) is based on the amino acid % of species conserved at each amino acid (0—no conservation, 1–100% conserved), placed on a 21 codon sliding window such that each site is added to 10 upstream and 10 downstream. The basic (blue, R/K/H), acidic (red, D/E), or S/T (orange) amino acid scores are calculated using only these amino acids for the same 21 codon window. Below is conservation of the amino acid locations (x-axis) for UniProt annotated domains/motifs, splice sites, posttranslational modifications (PTMs), and ELM annotated motifs (y-axis lists the number of annotations at each amino acid). (**B**) Extracted genomic variants from gnomAD (y-axis list allele frequency, bubble size is the number of unique variants), ClinVar (y-axis is the CADD functional score, color represents the type of variant), and Geno2MP variants (y-axis is the CADD score, bubble size the number of human phenotype profiles). Abbreviations—AA, amino acid; R, arginine; S, serine; T, threonine; LoF, loss-of-function; ELM, eukaryotic linear motif; HPO, human phenotype ontology.

**Figure 2 biology-12-00589-f002:**
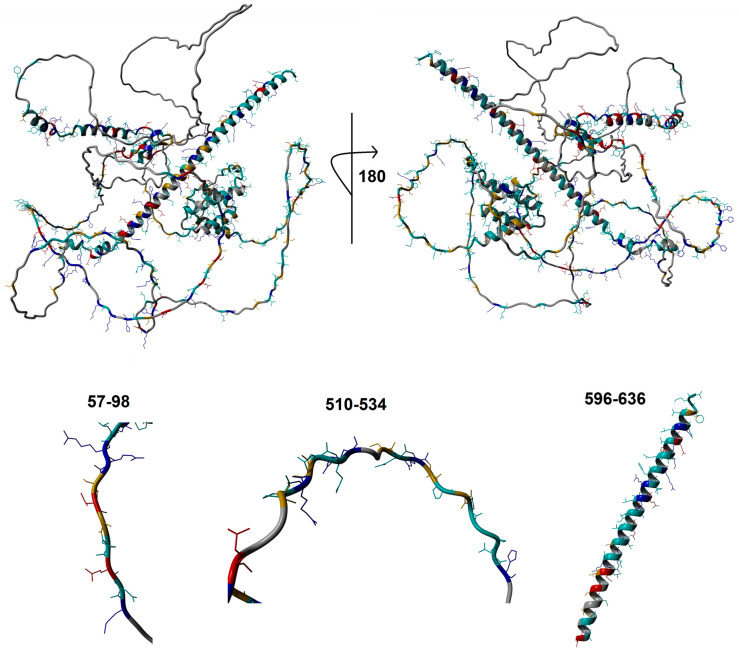
**Predicted WAC protein structure.** An AlphaFold protein model of WAC showing conserved amino acids colored as blue: polar basic; red: polar acidic; orange: S/T; cyan: all other conserved amino acids. Shown below are zoomed in view of several conserved motifs within WAC that fall within intrinsically disordered regions.

**Figure 3 biology-12-00589-f003:**
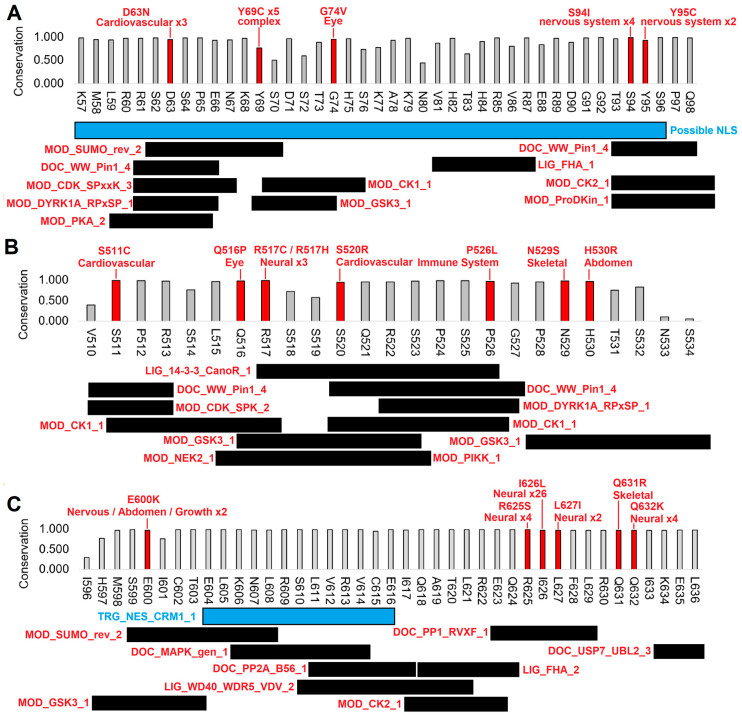
**Highly conserved motifs within WAC containing human variants connected to medical phenotypes.** Shown for the N-terminal nuclear localization motif (**A**), central 14-3-3 interaction motif (**B**), and the C-terminal nuclear export motif region of the coiled-coil domain (**C**). The conservation is based on the 364 amino acid sequences of vertebrate WAC (0—no conservation, 1–100% conserved). Those amino acids in red have a known Geno2MP variant with listed phenotypes. Those with multiple individuals with overlapping phenotypes are represented with an x number of individuals. Below each region are ELM-predicted motifs.

**Figure 4 biology-12-00589-f004:**
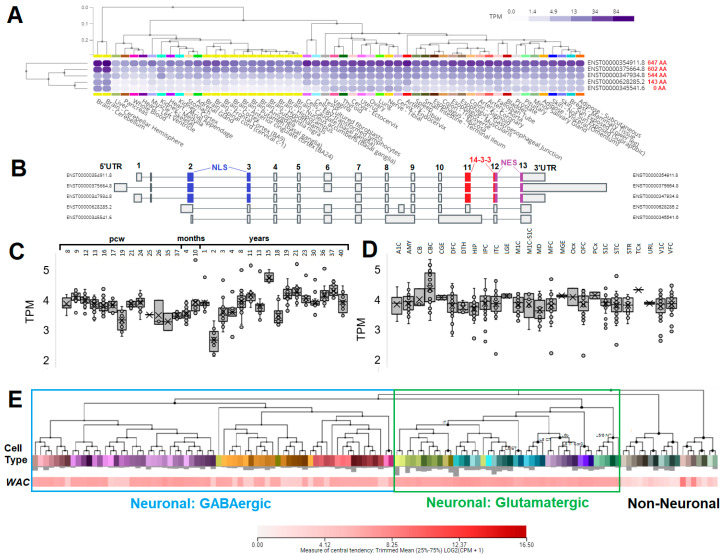
Analysis of WAC expression in various tissues, the brain developmental times, and within single-cell types of human brain. (**A**) GTEx expression of WAC isoforms in various human tissues. The darker blue color represents higher expression. Tissues are ranked in expression. In red next to each isoform number is the amino acid size of the protein. (**B**) Annotated exons of each isoform as shown in panel A. In blue are exons that code for the NLS, red the 14-3-3 interaction site, and magenta the NES. (**C**,**D**) The developmental transcriptome data of BrainSpan shown as box and whisker plots of age (**C**) or tissue type (**D**). Abbreviations include: pcw- post-coital weeks, A1C—primary auditory cortex (core), AMY—amygdaloid complex, CB—cerebellum, CBC—cerebellar cortex, CGE—caudal ganglionic eminence, DFC—dorsolateral prefrontal cortex, DTH—dorsal thalamus, HIP—hippocampus (hippocampal formation), IPC—posteroventral (inferior) parietal cortex, ITC—inferolateral temporal cortex (area TEv, area 20), LGE—lateral ganglionic eminence, M1C—primary motor cortex (area M1, area 4), M1C-S1C—primary motor-sensory cortex (samples), MD—mediodorsal nucleus of thalamus, MFC—anterior (rostral) cingulate (medial prefrontal) cortex, MGE—medial ganglionic eminence, Ocx—occipital neocortex, OFC—orbital frontal cortex, PCx—parietal neocortex, S1C—primary somatosensory cortex (area S1, areas 3,1,2), STC—posterior (caudal) superior temporal cortex (area 22c), STR—striatum, TCx—temporal neocortex, URL—upper (rostral) rhombic lip, V1C—primary visual cortex (striate cortex, area V1/17), VFC—ventrolateral prefrontal cortex, TPM—transcripts per million. (**E**) Single-cell expression from the Allen Brain Atlas Human MTG-10X (SEA-AD) data. Deeper red represents higher expression of WAC. Clustering shows grouped cells within labeled neuronal groups.

**Figure 5 biology-12-00589-f005:**
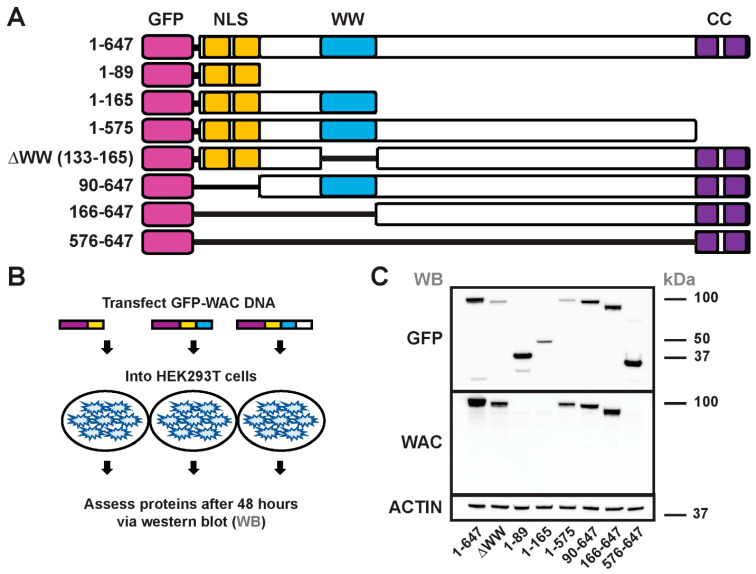
**Assessment of GFP-WAC deletion mutant proteins.** Schema showing the various GFP-WAC vectors generated to assess conserved protein domains (**A**). (**B**) GFP-WAC vectors were transfected into HEK293T cells to express proteins. (**C**) Western blots demonstrating expression of the various GFP-WAC proteins after 48 h. Top panel probed with an anti-GFP antibody and bottom panel probed with a commercial anti-WAC antibody. (kDa) kilodaltons.

**Figure 6 biology-12-00589-f006:**
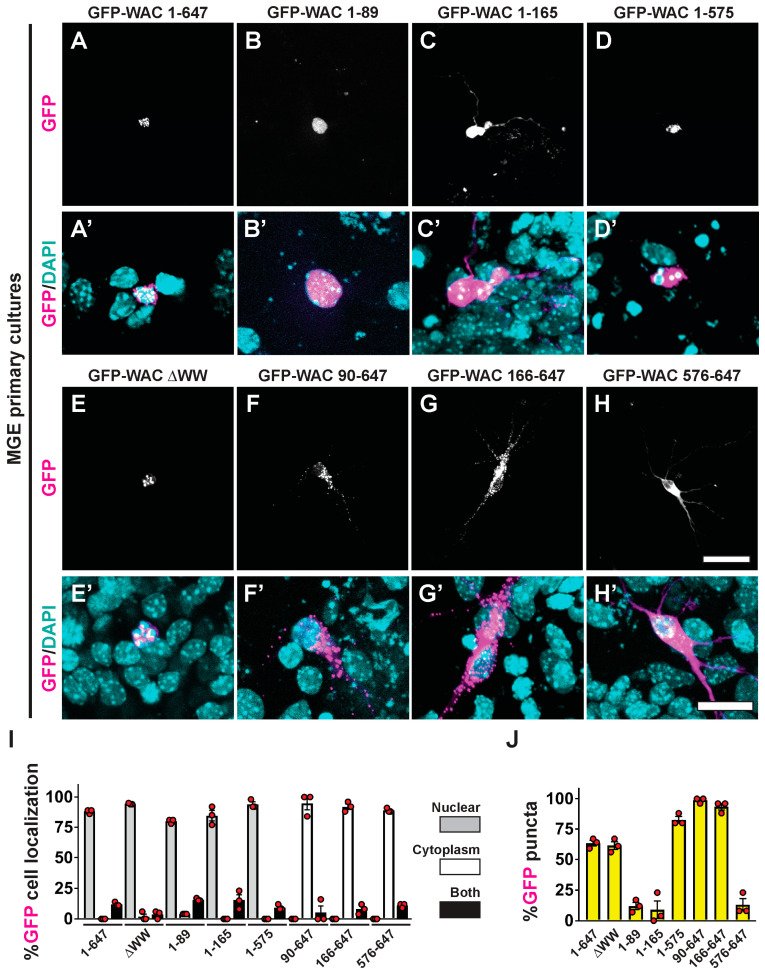
**Distinct cellular distribution by WAC’s conserved protein domains.** MGE primary neurons transfected with GFP-WAC fusion proteins were assessed for GFP localization (**A**–**H**) and merged with DAPI after five days in vitro (**A’**–**H’**). (**I**) Quantification of the proportion of GFP labeled cells showing nuclear and/or cytoplasmic distribution. (**J**) Quantification of the proportion of GFP-labeled cells with punctate localization. Scale bars in (**H**) = 40 µm for all top panels and (**H’**) = 20 µm for all bottom panels.

**Table 1 biology-12-00589-t001:** High ranking Geno2MP variants.

WAC Change	Conservation Score	CADD Score	PolyPhen2 Damaging	HPO Profiles	Homozygous Count	Abnormality Noted in HPO
S94I	98.87%	27.2	probably	4	3	nervous system
L426F	98.30%	31	probably	1	0	nervous system
R517H	98.58%	24.6	probably	1	0	nervous system
K466R	98.87%	24.1	probably	1	0	nervous system
P557A	97.45%	20.8	probably	1	0	nervous system
E600K	97.17%	22.1	probably	1	0	nervous system, abdomen
Y95C	92.63%	20.6	probably	4	0	nervous system, limbs, eyes
P464L	93.20%	34	probably	3	0	nervous system, genitourinary system, skeletal system
H530R	96.88%	23	probably	2	0	abdomen
K581R	98.02%	26.8	probably	2	0	cardiovascular system
S511C	98.58%	26.6	probably	1	0	cardiovascular system
S520R	95.47%	21.2	probably	1	0	cardiovascular system
G74V	95.47%	25.1	probably	1	0	eye
S449P	99.15%	24.7	probably	1	0	eye
Q516P	98.02%	20.6	probably	1	0	eye
T556M	97.45%	25.5	probably	1	0	genitourinary system
P526L	97.17%	34	probably	1	0	immune system, abdomen
S140I	99.43%	33	probably	2	0	limbs
H591R	98.02%	24.6	probably	2	0	limbs
P347S	98.58%	31	probably	2	0	skeletal system
Q631R	97.17%	25.2	probably	2	0	skeletal system

## Data Availability

A supplemental file with all details on the bioinformatics of the WAC sequence has been deposited at https://doi.org/10.6084/m9.figshare.22263538.v1. A description of columns can be found within the details of the file on FigShare. All other data presented in this study are available on request from the corresponding authors.

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
