# Peer review of "Structure-Function of the Human WAC Protein in GABAergic Neurons: Towards an Understanding of Autosomal Dominant DeSanto–Shinawi Syndrome"

_biology, 2023, doi:10.3390/biology12040589_

Round 1
Reviewer 1 Report
This work is very relevant since the authors decide to take the initiative to study one of the many proteins associated with a disease phenotype, but whose pathological function has not been clarified. Because the group of disease-associated proteins is so large, scientists interested in revealing their function are plagued by a lack of tools or workflow to help solve the main problem. Rudolph and collaborators present a very sturdy work about the WAC protein linked to a rare autosomal dominant disorder, the DeSanto-Shinawi syndrome. Using a variety of modern and state-of-the-art experimental tools and approaches, the authors reveal the most unique properties of WAC and its relevance in neurodegenerative disorders. However, it is necessary to complete the study with additional experiments to complement the results shown and clarify some of the statements that support the results to improve the quality of the study.
I-Major comments:
1- Methods:
1.1- For clarity, section “2.5. MGE primary cultures “should be before section “2.4. Immunofluorescence labeling and imaging”. When referring to "primary neurons", it is not clear if the authors meant MGE neurons. Although the methodology has been previously described; it would be beneficial to detail in the section "Primary cultures of MGE" a summary of how the MGE were obtained.
1.2- In the “2.4. Immunofluorescence Imaging and Labeling” section, there is no definition of how the quantification of the immunofluorescent signal was performed.
1.3- The section “2.7. Western Blot” used HEK293T cells. A detailed description of how the culture was maintained is lacking. It is necessary to include a paragraph with the missing information.
1.4- The section “2.6. Statistics and Cell Evaluations” lack detailed information on how the “cell assessment” was performed. This section should be reorganized as the last section of the Methods section.
2- Results and discussion:
2.1- Figure 1: This is an important figure that needs better representation by rearranging it. The alpha-fold structure of the WAC protein can be placed in a separate figure, where the detailed features can be better appreciated. The detailed description in lanes 198 to 207 should be incorporated with the alpha-fold structure. In particular, sequence alignment analysis showing conservation of polar residues should be shown.
Panels A and C should be placed together in a larger format, where the detailed information is better appreciated. Panel C needs to label the number of variants per phenotype. It is not clear what the vertical line in panel C means. Define all abbreviations: AA, amino acids; R, Arginine; S, serine; T, threonine; LoF, ELM, HPO, etc.
2.2- The explanation of the unique characteristics of the WAC protein (paragraph containing lanes 179 to 197) should be shown in a figure. It seemed that some of them were shown in Figure 2, but there is no mention of Figure 2 in the results section containing lanes 179-197.
2.3- In lane 192, the first mention of MAPK needs to be defined. Same for MED13 in lane 193, it would be appropriate to add a brief description (CDK8-kinase module component...).
2.4- The statement made in Lane 210 “there are loss-of-function variants (LoF), suggesting that truncation of the protein results in disease”, is not correct. The authors are implying that only nonsense variants result in loss of function and associated disease phenotype. Missense variants also result in loss of function associated with the disease phenotype. If the argument is to emphasize that, in the case of the WAC protein, only nonsense variants are linked to disease-related loss of function, that needs to be clarified.
2.5- When describing the results sections “Functional domain and motif annotations of WAC” and “Human clinical variants annotated to functional motifs of WAC”, it is necessary to specify which WAC isoform was used in the analysis.
2.6- The results section “Expression of WAC within human brain and neural cell types” should be expanded to describe detailed information relevant to the next section of results, related to panels D and E of figure 3.
2.7- Expression studies in HEK293T cells are useful to demonstrate the full expression of the different WAC constructs. The experiment that is missing, and that must be completed, is to perform an immunolocalization to demonstrate the distribution of the WAC protein in the different subcellular compartments by co-immunostaining with specific markers for the nucleus, ER and PM. Immunolocalization studies can be doubly confirmed with the WAC mutant constructs using the commercially available antibody that was used in the positive results. Similarly, these colocalization studies should be performed with the primary MGE neurons. The results shown in Figure 5 need to be complemented with specific markers that better show the colocalization of the WAC constructs and the different subcellular compartments. Also, the images shown to represent the results appear in low resolution. There is too much noise and graininess. Also, it is necessary to add arrows that point to what the authors are describing in the corresponding results section.
2.8- The TUNEL assay should be added to all cell culture studies to detect and measure whether there is any degree of cell death as a result of transfection and expression of WAC proteins.
2.9- One question that needs to be answered is to confirm that nuclear localization of WAC constructs promotes aggregate formation (“distinct punctate pattern in the nucleus”, lane 310) as a molecular mechanism for causing the associated DESSH phenotype. The observation of the “distinct punctate pattern in the nucleus” needs clarification and further experimentation to substantiate that the actual data “suggest that amino acids 166-575 may be responsible for this cellular pattern” (lane 340).
II-Minor comments:
1- Define first mention abbreviations. Wild type (WT) in line 101.
3- Specify the age of the embryos used for the primary culture (line 101).
4- Legend of figure 3: define TPM.
5- Legend of figure 5: Panels A-B are without scale bar. A panel has no label (D'?).
6- Clarify argument in lane 355-357: “To this end, the same WAC deletion mutants had similar localization patterns in HEK293T 356 cells (HCR and DV, data not shown).”
Author Response
Please find attached our responses as a word file.

Reviewer 2 Report
The authors explore the structure-function of the human WAC protein in GABAergic 2 neuron cells. The authors first analyzed the bioinformatics of WAC and identified potential functional motifs that were not annotated in the protein databank. Through clinic mutants comparison, the study showed connections between specific motifs and disease phenotypes. The authors use GFP fusion to investigate the cellular distribution of WAC protein containing various motifs and provided insights about the protein domain functionality. The work could provide some initial knowledge about the clinic phenotypes and protein potential function links and support the research for the field. Following are some questions:
1. In Figure 5, the authors imaged the transfected cells after 5 days, it didn't exclude the possibility of GFP-WAC proteins under degradation processing. The author should provide a explanation about the interpretation of the distribution without the effect of the degradation.
2. It would be best to include normal light field images in figure 5 to compare the whole cells and GFP/DAPI fluorescence.
3. In Figure 4, the authors used western blot to validate the GFP fusion protein. Does that mean the GFP fusion protein expression is so low that it's impossible to detect protein bands from GFP fluorescence?
Author Response

(The authors gave the same response as above.)

Round 2
Reviewer 1 Report
The authors responded substantially to most of the main criticisms of this reviewer without further experimentation. However, I still think that a better representation of Figures 1 and 2 would be beneficial for understanding the structural motifs of this unique protein. Instead of rearranging the Figures, as I suggested, the authors provided an online version of the protein structure and the Supplementary File with a detailed description of the protein motifs. While this is useful, for clarity and convenience for readers, having the full image in the document helps authors understand what is displayed. Unfortunately, not all future readers have full access to online resources. Being more inclusive is part of our scientific effort to spread knowledge. I recommend the authors to reorganize the figures and add a table with the most critical characteristics of the protein that would help to better describe the characteristics to which the authors refer.
Minor comments:
1-Figure 1: Panel C is not mentioned in the legend.
2-Figure 5: Panels A-D (A’-D') do not have scale bars. It seems they can't be the same scale bars used for E-H (E'-H').
Author Response
Please find attached our responses and edits.

Round 3
Reviewer 1 Report
The authors followed this reviewer's suggestion, which improved the comprehension of the manuscript and therefore has no further comments.